# Detection and Characterization of an H9N2 Influenza A Virus in the Egyptian Rousette Bat in Limpopo, South Africa

**DOI:** 10.3390/v15020498

**Published:** 2023-02-10

**Authors:** Rochelle Rademan, Marike Geldenhuys, Wanda Markotter

**Affiliations:** Centre for Viral Zoonoses, Department of Medical Virology, University of Pretoria, Pretoria 0001, South Africa

**Keywords:** influenza A virus, surveillance, Egyptian Rousette bat, South Africa, H9N2

## Abstract

In recent years, bats have been shown to host various novel bat-specific influenza viruses, including H17N10 and H18N11 in the Americas and the H9N2 subtype from Africa. *Rousettus aegyptiacus* (Egyptian Rousette bat) is recognized as a host species for diverse viral agents. This study focused on the molecular surveillance of a maternal colony in Limpopo, South Africa, between 2017–2018. A pan-influenza hemi-nested RT-PCR assay targeting the PB1 gene was established, and influenza A virus RNA was identified from one fecal sample out of 860 samples. Genome segments were recovered using segment-specific amplification combined with standard Sanger sequencing and Illumina unbiased sequencing. The identified influenza A virus was closely related to the H9N2 bat-influenza virus, confirming the circulation of this subtype among Egyptian fruit bat populations in Southern Africa. This bat H9N2 subtype contained amino acid residues associated with transmission and virulence in either mammalian or avian hosts, though it will likely require additional adaptations before spillover.

## 1. Introduction

Influenza A viruses are of public health importance due to the past pandemics they have caused and for their role as a re-emerging threat. Sporadic outbreaks and seasonal infections of influenza viruses can lead to significant morbidity and mortality in vulnerable populations [1]. Influenza A viruses are classified into subtypes based on their hemagglutinin (HA) and neuraminidase (NA) surface proteins. Before 2011, the ICTV recognized 16 HA and 9 NA types [2]; however, two new subtypes were identified in 2012 and 2013 [3,4]. The natural reservoir host for nearly all influenza A viruses (types H1–16; N1–9) is believed to be aquatic waterfowl [5,6], with the most recently described subtypes originating from bats [3,4,7]. The influenza A virus genome segments (negative sense RNA) consist of the polymerase basic 2 (PB2) gene, the PB1-, polymerase acidic (PA)-, the HA-, nucleoprotein (NP)-, the NA, matrix (M), and non-structural (NS) gene segments numbered from one to eight, respectively [8]. The broad host range of influenza A viruses can be attributed to the reassortment of its genome segments and gradual accumulation of mutations in the surface glycoproteins (HA and NA proteins) and the polymerase genes due to the error-prone RNA-dependent RNA-polymerase (RdRp). The M gene and the PB1 gene are selected as targets for molecular surveillance assays as they contain highly conserved regions. The PB1 gene served as the primary target for detecting novel influenza A viruses from bats [3,4,8], as mutations in the M gene have previously shown reduced efficacy of the assay and difficulty in detecting novel influenza subtypes [9,10].

H17N10 and H18N11 have only been detected in South American phyllostomid bats [3,4]. Despite being closely related, H17N10 and H18N11 have a high degree of divergence, indicating that a significant amount of time has passed since their diversification [4]. Influenza A viruses have also been identified in African bats in the Pteropodidae family. Serological evidence of an H9 subtype was identified among *Eidolon helvum* (Straw-colored fruit bat) from Ghana, with limited cross-reactivity to H8 and H12 [11]. Similarly, an H9N2 subtype was detected in the alimentary swabs of *Rousettus aegyptiacus* (Egyptian rousette bat) in Egypt [12]. This subtype formed a distinct lineage, separate from the other known bat influenza subtypes, suggesting that the virus may have originated from an avian influenza H9N2 subtype [12]. Serological evidence confirmed different roosts of *R. aegyptiacus* harbored the bat H9N2 virus [12]. Since these roosts likely form part of a larger metapopulation, the virus may circulate over a broader geographic range (much like metapopulation circulation, which has been suggested for Marburg viruses in Uganda) [13].

In Africa, the cave-dwelling *R. aegyptiacus* are recognized as viral hosts for multiple diverse pathogens, including the Marburg virus [14,15,16,17], coronaviruses [17,18,19,20], lyssaviruses (such as the rabies-related virus, Lagos bat virus) [21], various paramyxoviruses [17,22,23,24,25] including Sosuga virus [17,26], and influenza viruses [12]. *R. aegyptiacus* prefers tropical and sub-tropical regions with a distribution ranging from North Africa to sub-Saharan Africa, southwest Asia to Iran and Pakistan, including Cyprus [27,28,29]. It is a frugivorous bat that typically roosts near fruiting trees, orchards, or agricultural areas with a diet consisting of fruits, flowers, and sometimes even leaves [27,29]. Anthropogenic landscape changes, recreational activities, and traditional practices have increased animal-to-human contact, creating more spillover opportunities [30]. Increased influenza A virus detection and characterization from bat species have implicated bats as potential hosts for novel influenza viruses. In this study, we investigated the presence and diversity of influenza A viruses in an *R. aegyptiacus* maternal colony in Limpopo, South Africa.

## 2. Materials and Methods

### 2.1. Study Site and Biosafety

Samples were collected from *R. aegyptiacus* between June 2017 and December 2018 in the Matlapitsi cave in Limpopo (24°0′52.442′′ S, 30°9′58.967′′ E), South Africa. This cave is a maternal colony for *R. aegyptiacus*, with known co-roosting species, *Miniopterus* and *Rhinolophus*. The study site is situated in a rural area where other wildlife animals, free-roaming livestock, and agriculture are present. The samples were collected as part of a viral surveillance program of the Centre of Viral Zoonoses (CVZ) with DALRRD Section 20 approval and animal ethics approval from the University of Pretoria. All field and laboratory methods were performed following the guidelines and regulations of the institutional ethical approval boards. Samples were collected by vaccinated and trained individuals with the necessary biosafety precautions, including the Adflo™ battery-operated Powered Air Purifying Respirator (PAPR) System (3M, Johannesburg, South Africa), disposable Tyvek coveralls (DuPont, Johannesburg, South Africa), double latex gloves, rubber boots, and leather gloves. All equipment and clothing were disinfected with 10% bleach after sampling.

### 2.2. Sample Selection and Preparation

Fecal samples (*n* = 533) were collected from rock surfaces directly underneath the roosting *R. aegyptiacus* bats using wooden-handled swabs (Lasec, Johannesburg, South Africa), where three fecal boluses were smeared into one tube to represent a pool. Individual bats were captured using specifically placed harp traps (Faunatech Austbat, Victoria, Australia) near cave entrances. The harp traps used consisted of a large two-bank harp trap covering most of the cave entrance and a small three-bank trap positioned to a left-side opening. Oral swabs (*n* = 327) collecting saliva from the palate, tongue, and the inside of the cheeks from *R. aegyptiacus* were taken using a sterile cotton swab (Critical swab, VWR, Radnor, PEN, USA). All samples collected were kept in liquid nitrogen at the site and subsequently stored in a −80 °C freezer for long-term storage in the Center for Viral Zoonoses (CVZ) Biobank at the University of Pretoria. According to the manufacturer’s instructions, total RNA extraction of fecal samples occurred in a BSL-3 laboratory using the Quick-RNA™ MiniPrep Plus extraction kit (Zymo Research, Irvine, CA, USA). RNA extraction of oral samples was performed using the TRIzol extraction method (Invitrogen, Waltham, MA, USA) in a BSL-3 laboratory. The cDNA for the retrospective samples was prepared using the SuperScript™ IV Reverse Transcriptase kit (Thermo Fisher Scientific, Waltham, MA, USA). Briefly, an initial 13 μL primer-annealing reaction mixture was prepared using 10 mM dNTPs (Thermo Fisher Scientific, Waltham, MA, USA), 100 ng random hexamer primers (Integrated DNA Technologies, Coralville, IA, USA), nuclease-free water (Thermo Fisher Scientific, Waltham, MA, USA), and 8 μL of the extracted sample RNA and incubated at 65 °C for 5 min. After that, a reaction mixture containing 5× SuperScript™ IV First-strand Buffer (Thermo Fisher Scientific, Waltham, MA, USA), 0.1 M DTT (Thermo Fisher Scientific, Waltham, MA, USA), 40 U/μL Ribolock RNase Inhibitor (Thermo Fisher Scientific, Waltham, MA, USA), and 200 U/μL SuperScript™ IV Reverse Transcriptase (Thermo Fisher Scientific, Waltham, MA, USA) was added to the primer-annealing mixture, gently vortexed and centrifuged. The final 20 μL reaction mixture was placed into the Applied Biosystems SimpliAmp Thermal Cycler (Thermo Fisher Scientific, Waltham, MA, USA) and incubated at 23 °C for 10 min, followed by 55 °C for 50 min, and a final inactivation step at 80 °C for 10 min. A selection of the samples was subjected to an 18S rRNA internal control PCR assay to validate the RNA extraction methods used and the integrity of the extracted RNA (Appendix A).

### 2.3. Nucleic Acid Surveillance Assay

The PB1 gene segment contained highly conserved regions amongst influenza A viruses and was selected for primer design to develop a broadly applicable pan-influenza hemi-nested RT-PCR assay. A total of 60 complete PB1 gene sequences (Appendix A) of reference influenza A virus subtypes were retrieved from GenBank and aligned through BioEdit (v.7.0.5) [31]. Previously designed primers [32] targeting the PB1 gene region of influenza viruses were evaluated and modified for their use in a pan-influenza hemi-nested RT-PCR assay (Table 1) using AnnHyb (v.4.946) (http://bioinformatics.org/annhyb/) (accessed on 05 May 2019) and Integrated DNA technologies’ OligoAnalyzer (https://eu.idtdna.com/pages/tools/oligoanalyzer) (accessed on 30 April 2019).

Positive control transcript RNA of various influenza A virus subtypes were synthesized by GenScript (GenScript, Piscataway, NJ, USA) using a partial PB1 gene sequence. These controls included the H1N1pdm09 (NC_026435) and H3N2 (NC_007372) subtypes from a human origin, an H7N9 subtype (KP415772) detected from an avian host, and the bat-specific subtypes H9N2 (MH376908), H17N10 (CY103890), and H18N11 (CY125943). The constructs were propagated in competent JM109 *Escherichia coli* cells (Promega, Madison, WI, USA) using the pUC57 plasmid vector (GenScript, Piscataway, NJ, USA). The Megascript^®^ SP6 Transcription kit (Thermo Fisher Scientific, Waltham, MA, USA) was used together with the SP6 promoter sequence to generate RNA transcripts of the controls, which were quantified with the Qubit^®^ 3.0 Fluorometer and a Quant-iT^®^ RNA XR Assay Kit (Invitrogen, Thermo Fisher Scientific, Waltham, MA, USA) to estimate the detectable copy number. The cDNA synthesis was performed using the SuperScript™ IV Reverse Transcriptase kit (Thermo Fisher Scientific, Waltham, MA, USA), according to the manufacturer’s instructions, using 1 µL of extracted transcript RNA, 100 ng of random hexamer primers (Integrated DNA Technologies, Coralville, IA, USA), and 10 mM dNTPs (Thermo Fisher Scientific, Waltham, MA, USA). The molecular surveillance assay was optimized using the Taguchi method [33], with cDNA derived from the positive controls. This was followed by the first round of PCR for the pan-influenza hemi-nested RT-PCR assay, which consisted of 2 μL of randomly primed cDNA template with 1× DreamTaq buffer (Thermo Fisher Scientific, Waltham, MA, USA), 0.6 μM orthoAPB1936F1 forward primer (Metabion, Planegg, Germany), 0.2 μM orthoAPB11500R1/2 reverse primer (Metabion, Planegg, Germany), 0.3 mM dNTP mixture (Thermo Fisher Scientific, Waltham, MA, USA), 4.0 mM MgCl_2_ (Thermo Fisher Scientific, Waltham, MA, USA), and 1.25 U DreamTaq Polymerase (Thermo Fisher Scientific, Waltham, MA, USA), with 33.25 μL nuclease-free water (Ambion, Thermo Fisher Scientific, Waltham, MA, USA) to make a 50 ul reaction volume. The cycle conditions for the first round of the PCR assay consisted of an initial step at 94 °C for 1 min, 40 cycles with denaturation at 94 °C for 30 s, annealing at 38 °C for 30 s, and extension at 72 °C for 45 s, followed by a final extension cycle at 72 °C for 10 min.

The nested assay was prepared by combining 2 μL of the first-round template, 1× DreamTaq buffer (Thermo Fisher Scientific, Waltham, MA, USA), 0.6 μM orthoAPB11200F2 forward primer (Metabion, Planegg, Germany), 0.6 μM orthoAPB11500R1/2 reverse primer (Metabion, Planegg, Germany), 0.44 mM dNTP mixture (Thermo Fisher Scientific, Waltham, MA, USA), with a final concentration of 2.0 mM MgCl_2_ (Thermo Fisher Scientific, Waltham, MA, USA). A total of 1.25 U DreamTaq Polymerase (Thermo Fisher Scientific, Waltham, MA, USA), and 34.55 μL nuclease-free water (Ambion, Thermo Fisher Scientific, Waltham, MA, USA) was used for a 50 µL PCR reaction. The cycle conditions were similar for both rounds of the PCR assay, with changes in the number of cycles from 40 to 35 and the annealing temperature from 38 °C to 42 °C. The optimized assay was used for influenza A virus surveillance of RNA samples, followed by gel electrophoresis using a 1.5% agarose gel. All amplicons corresponding to 396 bp were excised and purified from the gel using the Zymoclean™ Gel DNsA Recovery kit (Zymo Research, Irvine, CA, USA), according to the manufacturer’s instructions. Purified amplicons were prepared for Sanger sequencing using the BigDye^®^ v3.1 Terminator Cycle Sequencing kit (Thermo Fisher Scientific, Waltham, MA, USA), followed by submission to the sequencing facility at the Faculty of Natural and Agricultural Sciences, the University of Pretoria, for sequencing using the ABI3500xl sequencer (Thermo Fisher Scientific, Waltham, MA, USA).

### 2.4. Full Genome Amplification

Full genome amplification was performed using universal IAV primers from Van den Hoecke et al. [34], targeting the conserved 5’- and 3′-non coding regions for each gene segment. This includes the common universal forward primer (F_CommonUni12; 5′-GCC GGA GCT CTG CAG ATA TCA GCA AAA GCA GG-3′) and the common universal reverse primer (R_CommonUni13; 5′-GCC GGA GCT CTG CAG ATA TCA GTA GAA ACA AGG-3′). Segment-specific primers were also designed (Appendix A) for a nested RT-PCR assay to compensate for low viral concentrations. Additional P-gene primers were designed and evaluated to ensure full internal coverage of these larger segments (Appendix A). The RNA of influenza-positive samples was used for cDNA synthesis with the SuperScript^®^ IV Reverse Transcriptase kit (Thermo Fisher Scientific, Waltham, MA, USA), using an initial 13 μL primer-annealing reaction mixture containing 2.5 μM forward universal primer (F_CommonUni12) (Metabion, Planegg, Germany), 10 mM dNTPs from Thermo Fisher Scientific (USA), as well as 11 μL template RNA [34]. The Phusion High Fidelity DNA Polymerase kit (New England Biolabs, Ipswich, MA, USA) was used for full genome amplifications. The reaction mixture contained a final 1× concentration of Phusion High-Fidelity Buffer (New England Biolabs, Ipswich, MA, USA), 0.2 mM dNTPs (Thermo Fisher Scientific, Waltham, MA, USA), 0.5 μM of the respective segment-specific forward and reverse primers (Metabion, Planegg, Germany), and 0.50 μL of 2 U Phusion High Fidelity DNA Polymerase (New England Biolabs, Ipswich, MA, USA). The assay conditions were adapted from Van Den Hoecke et al., [34]. They consisted of an initial denaturation cycle of 30 s at 98 °C, followed by 35 cycles of denaturation of 10 s at 98 °C and one annealing–extension step of 7.5 min at 72 °C, and was concluded with a final extension cycle of 10 min at 72 °C. The nested round of the RT-PCR assay utilized a similar 50 μL reaction mixture as described above with the segment-specific primers; however, the cycle conditions for the second round of amplification were optimized to include a separate annealing step at various temperatures, as determined by the primers and the size of the genome segments (Appendix A). Products of the appropriate sizes were excised, purified, and subjected to Sanger sequencing using the BigDye^®^ v3.1 Terminator Cycle Sequencing kit (Thermo Fisher Scientific, Waltham, MA, USA) and the ABI3500xl sequencer (Thermo Fisher Scientific, Waltham, MA, USA) at the Faculty of Natural and Agricultural Sciences, the University of Pretoria, sequencing facility. The sequenced influenza A virus gene segments were deposited to the National Center for Biotechnology Information (NCBI) GenBank database with accession numbers MZ073285 to MZ073290.

A segment-specific and unbiased sequencing method was combined for a final segment recovery approach of lower concentration segments within the remaining sample volume (HA and PB1 segments). The segment-specific approach required cDNA synthesis using 20 µM of the common universal IAV forward primer (F_CommonUni12) (Metabion, Planegg, Germany), 20 µM of both HA- and PB1-specific forward primers (Metabion, Planegg, Germany), with 10 mM dNTP mix, and 5 µL of RNA template. The final reaction method was placed in the thermal cycler for 45 min at 45 °C, followed by 80 °C for 10 min. The cDNA was used as a template in a DreamTaq PCR assay to amplify both the HA and PB1 genes using 10 µM of the common IAV universal primers and segment-specific primers, 1× DreamTaq buffer (Thermo Fisher Scientific, Waltham, MA, USA), 10 mM dNTP mixture, 2.5 mM MgCl_2_ (Thermo Fisher Scientific, Waltham, MA, USA), DreamTaq polymerase (Thermo Fisher Scientific, Waltham, MA, USA) at a final concentration of 1.25 U with 10 µL template and nuclease-free water. The thermal cycler conditions were initiated with a denaturation step at 94 °C for 1 min, followed by 30 cycles of denaturation at 94 °C for 30 s, annealing at 42 °C for 30 s, and extension for 4 min at 72 °C, and a final extension at 72 °C for 10 min.

For the unbiased approach, double-stranded cDNA was prepared using SISPA primers (100 µM SISPA K8N, Metabion, Planegg, Germany), the SuperScript IV kit, and a Klenow fragment (New England Biolabs, Ipswich, MA, USA) for second-strand synthesis [35]. First-strand synthesis was conducted according to the Superscript manufacturer’s instructions and followed by the addition of 0.21 U/µL RNase H (Thermo Fisher Scientific, Waltham, MA, USA), 312.5 U/µL Klenow fragment (New England Biolabs, Ipswich, MA, USA), and 10 x Klenow buffer for second-strand synthesis. The reaction mixture was incubated at 37 °C for 60 min, followed by 75 °C for 15 min using a thermal cycler. The cDNA was used as a template for amplification using a general DreamTaq PCR assay with 10 µM of SISPA K primer with the cycler conditions set as follows: first cycle at 98 °C for 30 s, followed by 40 cycles of 98 °C for 10 s, 55 °C for 30 s, and 72 °C for 1 min, and a final extension at 72 °C for 10 min. The PCR products of the segment-specific PCR and the unbiased SISPA PCR assay were cleaned-up using the DNA Clean and Concentrator^®^ kit (Zymo Research, Irvine, CA, USA), following the manufacturer’s protocol. The concentration of the final eluted DNA was determined using the Qubit 3.0 Fluorometer (Invitrogen, Thermo Fisher Scientific, Waltham, MA, USA) with a Quant-iT™ dsDNA HS Assay Kit (Invitrogen, Thermo Fisher Scientific, Waltham, MA, USA) and pooled together proportionally to a final concentration of 10 ng/µL. The DNA product was submitted to the Sequencing Core Facility of the National Institute for Communicable Diseases (NICD) for amplicon sequencing of the single library using Illumina’s NextSeq 550 platform. The assembled HA and PB1 gene sequences obtained from the data set were deposited to the NCBI GenBank database with accession numbers OQ216561 and OQ216562, respectively.

### 2.5. Sequence Assemblies, Phylogenetic Analysis, and Amino Acid Sequence Comparison

Phylogenetic analysis of the 396 bp PB1 gene region was performed with the CIPRES Science Gateway v.3.3 [36], using a ClustalW alignment containing multiple IAV PB1 sequences representing the influenza A virus diversity and sequences used for primer design. All sequences obtained following Sanger sequencing were edited and trimmed, followed by further analysis and comparison of the nucleotide and amino acid sequences. Following NextSeq sequencing, raw reads were subjected to quality control trimming and removal of host DNA using CLC Workbench v.22.0.2. The trimmed reads were used to create contigs and subsequently used for a de novo assembly as well as read mapping, using the bat-H9N2 genome from Egypt as reference [12]. Following segment genome recovery, phylogenetic analyses were conducted per segment with relevant reference sequences representing various subtypes, hosts, geographic origins, and all bat-borne influenza viruses. Model parameters were interpreted using JmodelTest [36] and used in a Bayesian phylogeny with BEAST v.1.10.4 [37] to construct Bayesian maximum clade credibility trees. Bayesian MCMC chains were set to 15 million states, sampling every 1000 steps, and convergence was confirmed via an effective sample size (ESS) of >200 [38]. Final Bayesian trees were calculated in Tree Annotator with a 10% burn-in [39]. Trees were viewed and edited in Figtree v.1.4.2 [40]. Complete segment-specific sequences were aligned to the protein reference sequences after translation using the ClustalW function in BioEdit v.7.0.5 [31] and after the open reading frame had been verified using ORFfinder (https://www.ncbi.nlm.nih.gov/orffinder) (accessed on 30 April 2021. MEGA-X [41] was used for protein matrix comparisons, which were created using the p-distance substitution model and converted to percentage similarity.

## 3. Results and Discussion

The developed degenerate-primer pan-influenza hemi-nested RT-PCR assay was evaluated for its sensitivity against synthesized positive control transcripts derived from various subtypes, geographic origins, and hosts. The assay was optimized using the Taguchi method and could ultimately detect between 1 × 10^1^ (H1N1), 1 × 10^2^ (H18N11, H7N9, H3N2, and H17N10), and 1 × 10^3^ (H9N2-bat) viral RNA copies. This assay was used for molecular surveillance of 860 samples (533 fecal samples and 327 oral swabs) collected from an *R. aegyptiacus* maternal colony between June 2017 and December 2018 in Limpopo, South Africa. A single positive sample from population-level fecal, UPE 556, was collected in January 2018. The size and the structure of the population fluctuate over time due to the recolonization of the roost for mating and parturition (or exodus of bats following a birth pulse). Specifically, the population size is at its largest over the birthing and lactation periods, between November and May in South Africa, due to the influx of new individuals into the population, and at its lowest in the winter period, from May to August [42].

The closest available relative of the PB1 amplicon sequence was identified as H9N2 from the same host in Egypt [12], which had a 91% nucleotide identity, thus confirming its distribution among the species in Southern Africa. Among the South African colony, January is primarily associated with lactation [29]. The timing coincides with results from Kandeil et al., [12] relating to the detection of H9N2 bat influenza viruses from *R. aegyptiacus* in Egypt. This suggests that the physiology and specific reproductive behavior of *R. aegyptiacus* may be involved in viral shedding. The detected proportion of positive samples (0.12%) from the studied bat population for the sampling period is low compared to the findings of Kandeil et al., [12] (8.7% detection). The assay’s sensitivity to the H9N2 subtype was 1 × 10^3^ viral RNA copies. The requirement of being capable of detecting the growing diversity of influenza viruses requires the use of degenerate primers, reducing the assay’s sensitivity. The sensitivity of the assay could also not be compared to the one-step conventional RT-PCR assay from Tong et al., 2012, nor the real-time RT-PCR assay from Kandeil et al., 2019, as the sensitivities were not specified. Studies have indicated a mean viral load of 1 × 10^4^–1 × 10^5^ viral RNA/μL from collected positive samples [43,44,45]. This viral load is also supported in rectal swabs of yellow-shouldered bats, as determined by a quantitative real-time PCR assay from Tong et al., 2012. Therefore, the pan-influenza hemi-nested RT-PCR assay developed here should be capable of detecting influenza viruses circulating in the bat colony. The integrity of the RNA and the success of RNA extraction for the retrospective samples were confirmed with an 18S rRNA PCR assay (method provided in Supplementary materials).

Full genome amplification and sequencing were performed for the detected influenza A virus, allowing the recovery of seven of the eight genome segments (using Sanger sequencing: PB2, PA, NS, N, NP, and M segments with NextSeq sequencing obtaining the HA segment). For NextSeq sequencing analysis, a targeted and unbiased approach was implemented. The targeted approach utilized primers targeting the conserved ends present on all influenza A segments as well as HA and PB1 segment-specific primers in order to amplify these segments. A SISPA-based approach was added to ensure the sequencing of the HA and PB1 segments in the event that they were highly diverse [35]. All enrichment approaches were pooled for library preparation and produced ~11.7 million raw reads following the sequencing of the single pooled library. After trimming, the reads were used for de novo assembly in CLC genomics workbench v22, where the complete HA segment was assembled as a contig. Approximately 2 million trimmed reads were mapped back to the HA contig and had an average read coverage of 165,076. Despite exhaustive attempts using both Sanger and NGS, only a partial sequence consisting of 1060 nucleotides could be obtained for the PB1 segment. This may be due to the low RNA concentration and limited sample volume. No contigs over 300 nucleotides could be assembled for the PB1 segment (only 6.36% of the trimmed reads contributed to these contigs). The resulting protein from the partial PB1 segment fell outside the well-known conserved regions. Despite this, the PB1 protein sequence was 97.98% identical to the bat-borne H9N2 first described by Kandeil et al., 2019. Phylogenies were constructed for the PB1 fragment and each complete gene segment to compare subtype clusters, confirming that the Bat/UPE556/RSA/2018 (H9N2) virus belongs to the same lineage as the Egyptian H9N2 bat-influenza virus (Appendix A). This lineage is more closely related to avian- and other mammalian influenza viruses than the bat-exclusive subtypes, H17N10 and H18N11.

The bat H9N2 (from Egypt and South Africa) is closely related to the other H9N2 subtypes; however, based on the phylogenetic trees of highly conserved genes (Appendix A), the bat-specific H9N2 may have preceded the diversification of avian and mammalian influenza viruses [12]. Molecular clock analysis by Ciminski et al., 2020 also supports the notion that bats may have acted as progenitor hosts for other IAVs, yet also suggests an equal probability of spillover into bats from an avian influenza precursor [46]. The former hypothesis is supported by phylogenetic analysis of the M gene (Appendix A) since the M1 protein is considered one of the slowest-evolving proteins [47,48]. The phylogenetic tree for the NS segment (Appendix A) introduces an additional cluster of avian sequences between the new world bat viruses and the H9N2 bat-influenza viruses, which could indicate a possible early reassortment event that preceded the diversification of the gene segment.

The pairwise identity and amino acid changes for each segment of the Bat/UPE556/RSA/2018 (H9N2) were determined in reference to H9N2 subtypes from various avian and mammalian hosts, including the Egyptian H9N2 bat virus (Table 2). Despite variations in the nucleotide sequences between the H9N2 bat viruses and other H9N2s, the internal gene proteins indicated an amino acid identity ranging from 81.27% to 91.30%, except for the NS1 protein, with a peak amino acid identity at 72.60% for an ostrich-borne H9N2. The bat-borne HA and NA were quite diverse from other H9N2, with the highest amino acid identity (72.99% and 65.59%, respectively) relating to an H9N2 virus from swine. As expected, the M1 protein was highly conserved among H9N2 viruses, with an amino acid identity ranging from 90.48% to 95.63%.

Comparison of the M2- and NS1-proteins of the bat-borne H9N2 viruses specify a greater nucleotide identity than protein similarity, reflecting nucleotide differences at the 3rd position of the codon and resulting in greater amino acid variation between the two sequences. Nucleotide insertions within the highly variable stalk region of the Bat/UPE556/RSA/2018 (H9N2) NA-protein gene were observed compared to the H9N2 bat virus (Figure 1).

The Bat/UPE556/RSA/2018 (H9N2) HA segment phylogenetically clusters with the Egyptian H9N2 bat-borne influenza virus (sharing 91.09% nucleotide identity) (Figure 2). Despite the amino acid variations observed between the Bat/UPE556/RSA/2018 (H9N2) HA segment and the Egyptian H9N2 bat-influenza, key residues were conserved at the receptor binding site (RBS) of the HA segment (Appendix A). These residues are N166, H191, E197, Q198, G232, Q234, G235, R236, and Q399, where Q234 is associated with the avian α-2,3-sialic acid receptor [12]. The HA protein sequence also revealed an identical cleavage motif shared between the two bat-borne H9N2 viruses, as indicated in Appendix A, as well as conserved glycosylation sites at residue positions 298 (NSTL) and 305 (NISK) [12].

The NA-protein did not contain the residue (V275) associated with the α-2,6-sialic acid receptor (mammalian) [49,50]. The head domain of the Bat/UPE556/RSA/2018 (H9N2) NA amino acid sequence contained four glycosylation motifs (N-X-S/T) at position 143 (NGT), 197 (NAT), 231 (NGT), and 447 (NGT), with two highly conserved glycosylation signals in the stalk region, NIT at position 51, and NNT at position 64. The N2 head domain consisted of both a catalytic and antigenic site and contained most of the residue variations observed between bat/UPE556/RSA/2018 (H9N2) and its Egyptian counterpart. However, only one amino acid change (K328R) was observed within the antigenic site on the protein’s surface. Important to note is that the Bat/UPE556/RSA/2018 (H9N2) did not contain any molecular markers that confer resistance to neuraminidase inhibitors, such as V119, K136, K292, K222, N246, Y274, and S294. The Bat/UPE556/RSA/2018 (H9N2) virus did, however, display early adaptation markers for resistance against intrinsic antiviral activity [51]. Protein analysis of the Bat/UPE556/RSA/2018 (H9N2) virus revealed multiple residues associated with mammalian transmission and virulence (Table 3), despite the preference for avian, the α-2,3-sialic acid receptor from both the HA and NA proteins and selected residues was indicative of an avian host (such as K702 and E627) in the PB2 segment. However, most mammalian species, including humans, possess both receptor types in their respiratory tracts. Additionally, it was shown that some bat species display abundant co-expression of both receptor types [52], implicating bats as potential intermediate hosts.

The H9N2 subtype has been highlighted as a potential zoonotic agent as it can infect several mammalian species, including humans, dogs, minks, pigs, and horses [53,54,55,56,57], and has had a significant economic impact on the poultry industry [58,59,60,61]. The subtype has also been reported as able to infect various bat cell lines. It can also recognize avian and mammalian receptor linkage types [62], increasing the possibility of reassortment and further adaptation to other mammalian hosts. However, the H9N2 bat viruses likely require additional adaptations before being capable of zoonotic spread, as this virus is quite distinct from H9N2 viruses previously detected in chickens, ostriches, and humans. However, the zoonotic risk is ever-present with increased contact between humans and wildlife. The key residues indicated in Table 3 are most likely essential for the successful infection and spread of the H9N2 bat subtype among the *R. aegyptiacus* populations across various widespread geographic regions. Additional information regarding residues and signal motifs not directly involved in transmission and virulence within the Bat/UPE556/RSA/2018 (H9N2) segments is available in Appendix A.

**Table 3 viruses-15-00498-t003:** Important residues for mammalian transmission, virulence, and resistance that are present in the Bat/UPE556/RSA/2018 viral protein sequences.

IAV Proteins	Importance	Residues Present	References
**PB2**	Mammalian transmission and virulence	S199, V504, N701	[63,64,65]
**PA**	Mammalian transmission and virulence	N55, Y241, S404, V127, L550, L672	[64,66]
**PA-X**	Enhanced host shutoff	P28, S65, K195, K198, K202, K206, and the 233–252 end of the PA-X protein	[67,68,69]
	Increased virulence	R195K	[70]
**NP**	Early adaptation markers for Mx-protein antiviral resistance	K305, K351	[71]
**M2**	Increased virulence	P69	[72]

## 4. Concluding Remarks

The A/Rousettus aegyptiacus/South Africa/UPE556/2018 (H9N2) influenza virus was detected from a bat colony in a rural area of the Ga Mafefe, with free-roaming domestic animals, livestock, and wildlife animals with frequent human agricultural activities in the region. The characterized H9N2 bat virus contained amino acid residues associated with mammalian transmission and virulence, as well as avian-specific residues. The data generated in this study can be used as a comparison for future surveillance to monitor the prevalence and evolution of this virus in Southern African *R. aegyptiacus* populations and to determine zoonotic risk. This study has indicated the importance of surveillance and genome characterization to identify potential risks to public health, especially since some internal gene segments of H9N2 viruses are thought to contribute to the successful zoonotic transmission of avian influenza viruses [73].

Future surveillance studies should consider optimizing the available detection methods to include novel and diverse influenza A viruses and improve the detection limits of the assays. Subsequent surveillance efforts can also be directed to other *R. aegyptiacus* populations in South Africa, as these bats typically form large metapopulations covering broad geographic areas and include multiple roosts. Longitudinal sampling should be considered with the availability of demographic data of the bats to further our understanding of the H9N2 bat-influenza epidemiology and excretion dynamics. Furthermore, the seroprevalence of this influenza A virus in bats, as well as human populations considered high risk for zoonotic transmission (based on frequent contact with bat populations or in close proximity of a roost) can be conducted in conjunction with molecular detection to determine accurate distribution and epidemiology of the virus.

## Figures and Tables

**Figure 1 viruses-15-00498-f001:**
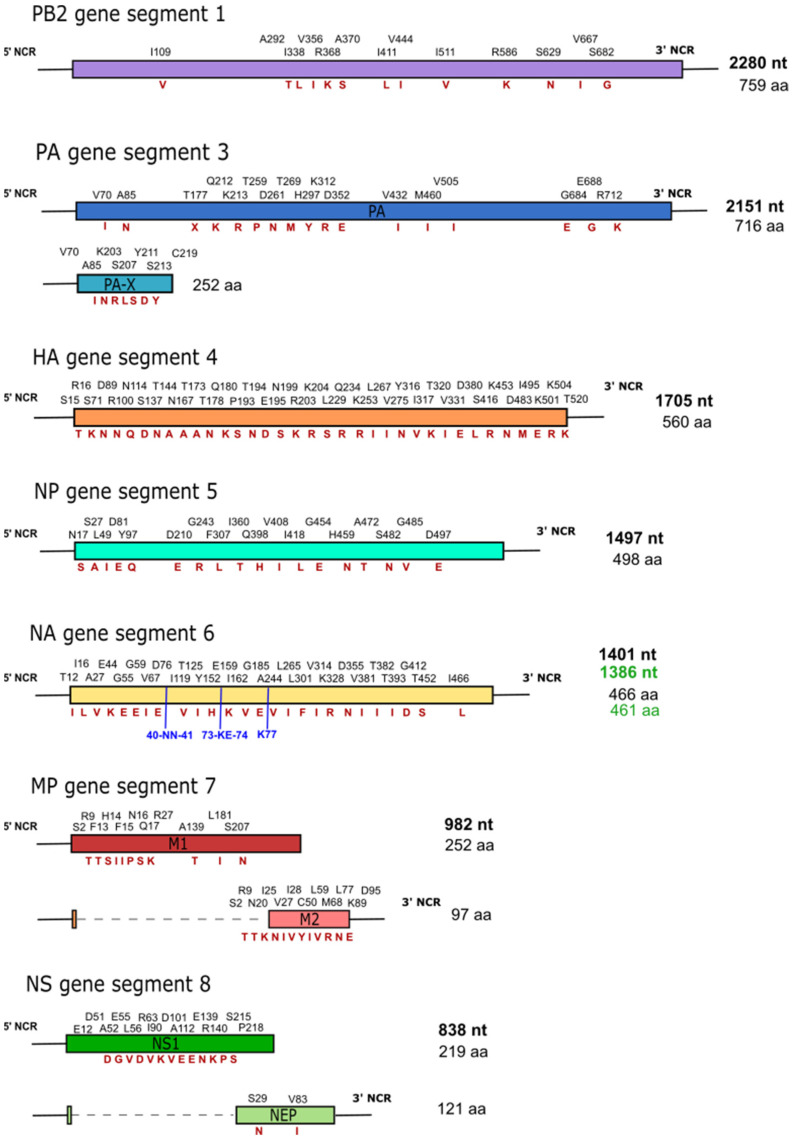
Representation of the Bat/UPE556/RSA/2018 (H9N2) virus segments (Accession: MZ073285-MZ073290, HA: OQ216561) and the residue variations when compared to the Egyptian H9N2 bat-virus (MH376902-MH376909). The residues and their relative positions, as seen for the Egyptian H9N2 bat virus, are indicated above each segment, with the variations in red. The nucleotide length of the Bat/UPE556/RSA/2018 (H9N2) (indicated in bold) and the amino acid length are shown on the right of each segment. The black nucleotide- and amino acid length shown represents the Bat/UPE556/RSA/2018 (H9N2) virus, with the Egyptian H9N2 bat-virus indicated in green due to the size differences observed for the NA segment. Additionally, blue residues with their respective positions indicate insertions, as seen for the NA-segment of the Bat/UPE556/RSA/2018 (H9N2) virus.

**Figure 2 viruses-15-00498-f002:**
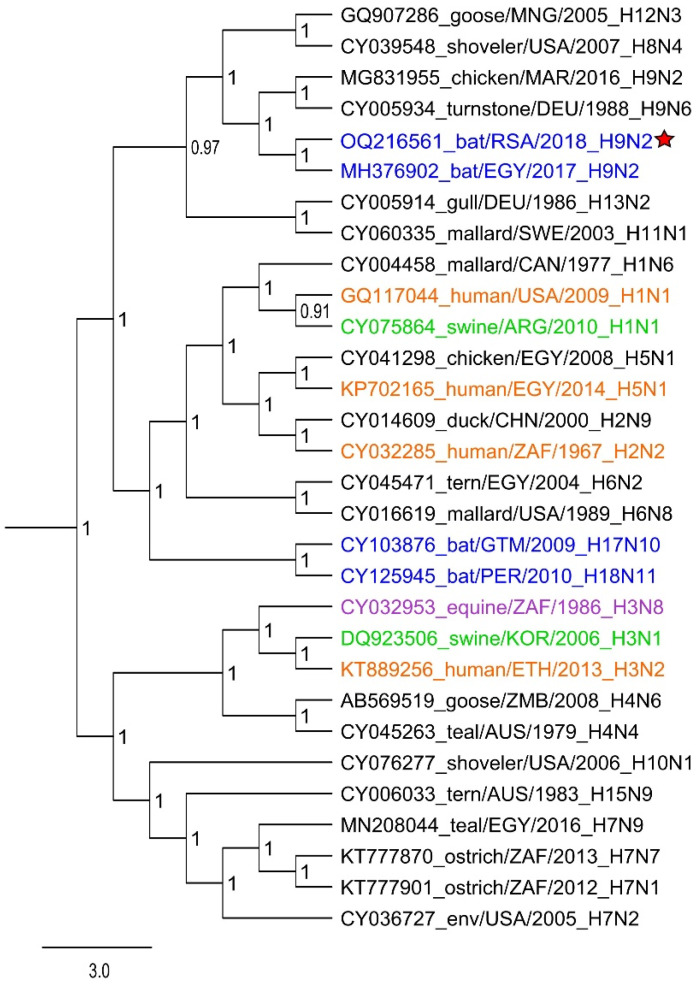
Bayesian phylogenetic analysis of the HA-segment of the Bat/UPE556/RSA/2018 influenza virus (accession: OQ216561) within the H9 clade as indicated by the red star graphic. BEAST (v.1.10.4) was employed using the GTR + I + G substitution model (MCMC chain set at 15 million). All influenza A viruses of mammalian origin have been highlighted with color, with viral sequences from bats indicated in blue, viruses originating from humans highlighted in orange, viruses with a swine host indicated in green, and viruses detected in equine hosts indicated in purple.

**Table 1 viruses-15-00498-t001:** Primers targeting the polymerase basic 1 (PB1) gene segment for a pan-influenza hemi-nested RT-PCR assay.

Primer	Sequence 5′-3’
**First round RT-PCR**	**Expected amplicon size: 699 bp**
orthoA PB1 936 F1	GGR GAC AAY ACM AAR TGG AAT G
orthoA PB1 1500 R1/2	GTT KAT CAT RTT GKT YTT KAT YAC TG
**Hemi-nested RT-PCR**	**Expected amplicon size: 396 bp**
orthoA PB1 1200 F2	CCW GGR ATG ATG ATG GGN ATG TTC

Primers are named according to the virus target, genome segment, approximate position in the reference genome (H1N1, accession: NC_026435.1), and primer orientation. Degenerate nucleotides are as follows: R = A/G, Y = T/C, M = A/C, K = T/G, W = A/T, N = any nucleotide.

**Table 2 viruses-15-00498-t002:** Pairwise comparison of the Bat/UPE556/RSA/2018 (H9N2) influenza A virus genome segments and viral protein sequences with other H9N2 subtypes from various host origins. The nucleotide (nt) and amino acid (a/a) identity is indicated as a percentage, with values above 90 highlighted in bold.

H9N2 Hosts	Bat		Human		Swine		Chicken		Duck		Ostrich	
Accessions	MH376902-09		NC_004905-12		KX421147-54		ON374567-73		MW531640-47		GQ404721-27	
Sequence	nt	a/a	nt	a/a	nt	a/a	nt	a/a	nt	a/a	nt	a/a
PB2	**95.04**	**98.29**	78.03	**90.25**	77.24	89.72	76.93	**90.25**	78.42	89.59	77.98	89.99
PB1 *	**94.12**	**97.97**	76.88	89.57	80.06	89.86	77.65	**91.01**	78.61	**91.30**	76.97	**90.72**
PA	**94.47**	**97.76**	78.24	88.81	77.96	87.41	79.03	88.25	78.70	87.96	79.17	89.51
PA-X	**94.47**	**97.22**	78.24	82.14	77.96	81.35	79.03	81.75	78.70	81.27	79.17	82.94
HA	**91.07**	**93.74**	68.04	70.84	68.69	72.99	67.86	69.77	67.14	68.87	68.81	72.45
NP	**94.05**	**96.36**	77.96	88.76	78.29	89.36	79.29	89.36	78.96	89.16	78.09	89.76
NA	**93.94**	**94.36**	65.95	64.79	66.31	65.59	66.17	65.59	66.31	64.52	65.60	64.73
M1	**93.69**	**95.63**	79.53	**92.46**	80.04	**94.05**	77.49	**90.48**	78.51	**90.48**	79.12	**93.65**
M2	**93.69**	84.54	79.53	73.20	80.04	73.20	77.49	68.04	78.51	69.07	79.12	72.16
NS1	**95.17**	**94.06**	74.85	70.78	76.49	71.23	76.14	70.32	75.32	68.95	76.00	72.60
NS2	**95.17**	**98.35**	74.85	86.78	76.49	87.60	76.14	85.95	75.32	86.78	76.00	88.43

* PB1 partial gene and protein sequence.

## Data Availability

Data presented in this study is openly available at ncbi.nlm.nih.gov (accessed on 11 January 2023).

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
