# Peer review of "Detection and Characterization of an H9N2 Influenza A Virus in the Egyptian Rousette Bat in Limpopo, South Africa"

_viruses, 2023, doi:10.3390/v15020498_

Round 1

Reviewer 1 Report

In their manuscript, Rademan et. al. present a detection and molecular characterization of influenza A/H9N2 virus in Egyptian Rosette bat in South Africa. Overall, the paper is well written and clear. However, several issues need to be addressed before the paper is acceptable for publication.

-       Consider adding more details about the ecology of bats in South Africa and human-bats interactions in introduction section

-       Was any further assessment done with the genetic data such as estimation of time to most recent common ancestor or any sort of phylogeographical assessment? These would be ideal to further the findings in the manuscript.

-       Did you carry out viral isolation trials in ECE or cells?

-       In the last paragraph, the authors mention limitations of the study (of which there are many), but they are not specifically detailed. Please consider expanding the discussion of these limitations throughout the manuscript.

-       Could you please add more details about vaccination of individuals that collected samples, type of PAPR, ethical approval no.?

-       Authors collected fecal samples from rock surfaces of cave that included several species of bats. Could you please adding the method that applied to recognize the host of fecal samples?

-       Rational of developing new method for detection of pan-influenza A should be included. if authors tested samples using previous method, it would be considerable. Additionally, the newly created hemi-nested RT-PCR assay needs to be examined for a variety of influenza A virus strains from various hosts.

-       For NGS of HA and M segments, more data about reads should be included.

-       Table 2,3  and Tables in supplementary file, please add a column for segments and other for viral proteins. For example, PA-X is not a segment.

-       Figure 2 needs to add coloring codes.

-       References needs more revision. For example, reference 46 line 373, reference 8 and 53   in table 3.

-       H and N in Table S1 should be HA and NA , NEP ….NS

-        

Author Response

Reviewer comment: In their manuscript, Rademan et. al. presents a detection and molecular characterization of influenza A/H9N2 virus in Egyptian Rosette bat in South Africa. Overall, the paper is well written and clear. However, several issues need to be addressed before the paper is acceptable for publication.

Author response: We would like to thank the reviewer for their time and feedback to improve the manuscript for publication, as well as for considerations for future and similar projects. All comments were carefully considered and additions to the manuscript were made as necessary. Grammarly was used to improve the language and style and citations have been updated accordingly.

Reviewer comment: Consider adding more details about the ecology of bats in South Africa and human-bats interactions in introduction section.

Author response: We thank the reviewer and have added a short section on ecology of the Egyptian Rousette bat to the introduction (as this is the bat species under investigation), as well as some additional information regarding the human-bat interactions.

Reviewer comment: Was any further assessment done with the genetic data such as estimation of time to most recent common ancestor or any sort of phylogeographical assessment? These would be ideal to further the findings in the manuscript.

Author response: Thank you for the comment and the viable suggestion. We did not estimate time to most recent common ancestor due to a paper published by Ciminski et al., 2020 that included time-calibrated phylogenies for the bat-borne H9N2, H17N10, and H18N11 subtypes in comparison with conventional influenza A viruses in which they investigated the HA gene segment and the PB1 gene segment as an internal gene representative. The purpose of this manuscript is to highlight the detection of this virus within an expanding geographical range, as it was previously only identified in Egypt - which was most likely due to limited surveillance. For this paper, we also wanted to share the genomic similarities and differences between the two viruses that can be used for other downstream applications and future projects. We do not believe that performing our own time-calibration phylogenetic analysis will contribute more than the paper by Ciminski et al., 2020. We believe that the phylogeographical assessment is a great suggestion, and will be applied in future work as we would aim to include a few more sequences into the analysis for better support.

Reviewer comment: Did you carry out viral isolation trials in ECE or cells?

Author response: We thank the reviewer for the comment. Unfortunately, we were unable to attempt virus isolation due to limited sample volume. Due to already limited sample material, we prioritized obtaining as much sequence information as possible. Pending future surveillance results, virus isolation will be attempted.

Reviewer comment: In the last paragraph, the authors mention limitations of the study (of which there are many), but they are not specifically detailed. Please consider expanding the discussion of these limitations throughout the manuscript.

Author response: Thank you for your time to review the manuscript and for the comment. We used the final paragraph of the manuscript to highlight the limitations of the study once more with ways to improve on the work that was done and included suggestions for future studies. We have included limitations and discussions around these issues within the general manuscript, also detailing efforts to overcome the limitations; as with the text from lines 296-310: “The detected proportion of positive samples (0.12%) from the studied bat population for the sampling period is low compared to the findings of Kandeil et al., 2019 (8.7% detection). The assay's sensitivity to the H9N2 subtype was 1x103 viral RNA copies. The requirement of being capable of detecting the growing diversity of influenza viruses requires the use of degenerate primers, reducing the assay's sensitivity. The sensitivity of the assay could also not be compared to the one-step conventional RT-PCR assay from Tong et al., 2012, nor the real-time RT-PCR assay from Kandeil et al., 2019, as the sensitivities were not specified. Studies have indicated a mean viral load of 1x104-1x105 viral RNA/μl from collected positive samples [42-44]. This viral load is also supported in rectal swabs of yellow-shouldered bats, as determined by a quantitative real-time PCR assay from Tong et al., 2012. Therefore, the pan-influenza hemi-nested RT-PCR assay developed here should be capable of detecting influenza viruses circulating in the bat colony. The integrity of the RNA and the success of RNA extraction for the retrospective samples were confirmed with an 18S rRNA PCR assay (method provided in supplementary materials).”

Reviewer comment: Could you please add more details about vaccination of individuals that collected samples, type of PAPR, ethical approval no.?

Author response: We thank the reviewer for the comment. The type of PAPR has been added to the manuscript. The ethical approval number, as well as additional permits and approvals are available in the institutional review board statement section according to the journal’s specifications as this improves readability of the manuscript. All individuals that participate in field work are routinely vaccinated against Rabies virus (Rabipor from Chiron Behring), Hepatitis A and B virus (Twinrex from GSK), Tetanus (Tetavax from Sanofi), Measles, Mumps, and Rubella (MMR vaccine, OMZYTA from MSD), and Meningitis (Menactra from Sanofi) with all booster doses completed and titres regularly monitored. Since the COVID-19 pandemic, all individuals have also been vaccinated against SARS-CoV-2 using either the Pfizer vaccine or Johnsons and Johnsons vaccine. It is also recommended to obtain a seasonal flu vaccine. This information was not included in the manuscript as it is considered general practice (and is included as an ethics requirement) and out of consideration for the manuscript’s readability, however the details will be made available when requested.

Reviewer comment: Authors collected fecal samples from rock surfaces of cave that included several species of bats. Could you please adding the method that applied to recognize the host of fecal samples?

Author response: We thank the reviewer for the interesting comment. The roosting site included the frugivorous Egyptian rousette bat, Rhinolophus bats (insectivorous), and Miniopterus bats (insectivorous). These two insectivorous bat populations tend to produce well-defined, small faecal pellets that can easily be distinguished from the larger mucilaginous faecal boluses of the Egyptian rousette bat. These boluses are collected with swabs as forceps would make it difficult to pick up. Insectivorous bat fecal pellets cannot be collected with swabs, unless it is used to push the pellet into a collection tube- one thus requires forceps. The clear difference in sample type also provides confidence in the origin of the samples. Though these bats may co-roost, each genus has a preferential site within the cave itself, although this is not always well defined. We have noticed that the Rhinolophus bats tend to roost further away in smaller chambers, whereas the Egyptian rousette bat prefers the main chamber close to the cave opening (most likely due to the seasonal aggregation of large colony numbers). The miniopterids tend to cluster together deeper into the cave. Therefore, the collection of faecal boluses from underneath roosting Egyptian rousette bats are attainable with clear species differentiation. We did not add any additional information to the manuscript as we already stated that the faecal samples were collected from underneath the roosting Egyptian rousette bats. We did, however, restructure the sentence slightly to reduce ambiguity.

Reviewer comment: Rational of developing new method for detection of pan-influenza A should be included. if authors tested samples using previous method, it would be considerable. Additionally, the newly created hemi-nested RT-PCR assay needs to be examined for a variety of influenza A virus strains from various hosts.

Author response: We thank the reviewer for their comment. We opted to establish a new protocol to allow detection of a large genetic diversity of influenza A viruses. We also attempted to contact authors of other bat surveillance studies to share primers (when primers were indicated to be ‘shared upon request’) and never received feedback. This protocol makes use of a conventional PCR method and allows for a greater amplicon size than desired for real-time assays. We preferred this option for improved preliminary analysis of the amplicon sequence to determine follow-up protocols. The assay we established also consists of two rounds of amplification using hemi-nested primers to ensure the detection of low RNA concentration samples. We also made use of in-house designed primers from Geldenhuys et al., 2018 as this data was readily available at the start of the study. The amplicon region in Tong et al., 2012 and Tong et al., 2013 was limited to 250 bp, and the primers were not provided, nor were they available for consideration. The assay described by Kandeil et al., 2019 contained an amplicon size of 403 bp; however, the same primer set was used for two rounds of the RT-PCR assay, which may introduce amplification of non-specific products. We did not want to risk non-specific amplification to that extent since we were testing environmentally collected faecal material. We examined our hemi-nested RT-PCR assay using synthetically generated controls from various influenza A virus subtypes consisting of H1N1pdm09 (human), H3N2 (human), H7N9 (avian), and the bat-specific subtypes H9N2, H17N10, and H18N11. This information was detailed in the materials and methods section:
“Positive control transcript RNA of various influenza A virus subtypes (H1N1pdm09, H3N2, H7N9, and the bat-specific subtypes H9N2, H17N10, and H18N11) were synthesized by GenScript (GenScript, Piscataway, NJ, USA) using a partial PB1 gene sequence”. In response to the reviewer’s comment, we have added the host origin to each positive control as well as their respective accession numbers to help inform those unfamiliar with each subtype: lines132-137 “Positive control transcript RNA of various influenza A virus subtypes were synthesized by GenScript (GenScript, Piscataway, NJ, USA) using a partial PB1 gene sequence. These controls included the H1N1pdm09 (NC_026435) and H3N2 (NC_007372) subtypes from human origin, an H7N9 subtype (KP415772) detected from an avian host, and the bat-specific subtypes H9N2 (MH376908), H17N10 (CY103890), and H18N11 (CY125943).”.

Reviewer comment: For NGS of HA and M segments, more data about reads should be included.

Author response: Thank you for the comment to improve the detail in the manuscript. We assume the reviewer means the HA and PB1 segments as the M segment was sequenced with sanger sequencing. We have included more data about the reads for the HA and PB1 segments that were subjected to NGS: Lines: 311-330: “Full genome amplification and sequencing were performed for the detected influenza A virus, allowing the recovery of seven of the eight genome segments (using Sanger sequencing: PB2, PA, NS, N, NP, and M segments with NextSeq sequencing obtaining the HA segment). For NextSeq sequencing analysis a targeted and unbiased approach was implemented. The targeted approach utilized primers to the conserved ends present on all influenza A segments as well as HA and PB1 segment-specific primers in order to amplify these segments. A SISPA-based approach was added to ensure sequencing of the HA and PB1 segments in the event that they were highly diverse. All enrichment approaches were pooled for library preparation and produced ~11.7 million raw reads following sequencing of the single pooled library. After trimming, the reads were used for de novo assembly in CLC genomics workbench v22, with the complete HA segment was assembled as a contig. Approximately ~2 million trimmed reads were mapped back to the HA contig and had an average read coverage of 165,076. Despite exhaustive attempts using both Sanger and NGS, only a partial sequence consisting of 1060 nucleotides could be obtained for the PB1 segment. This may be due to the low RNA concentration and limited sample volume. No contigs over 300 nucleotides could be assembled for the PB1 segment (only 6.36% of the trimmed reads contributed to these contigs).”

Reviewer comment: Table 2,3 and Tables in supplementary file, please add a column for segments and other for viral proteins. For example, PA-X is not a segment.

Author response: Thank you for the comment. Considering Table 2 in the manuscript, the description of the table has been updated, as well as one of the headings to include both the segments and their accompanying viral proteins. The table heading for Table 3 has been updated as requested.

Reviewer comment: Figure 2 needs to add coloring codes.

Author response:  We thank the reviewer for their comment as it greatly improves the overall quality of the figure. We have added the colouring codes to the figure caption. Lines: 392-394: “All influenza A viruses of mammalian origin have been highlighted with colour, with viral sequences from bats indicated in blue, viruses originating from humans highlighted in orange, viruses with a swine host indicated in green, and viruses detected in equine hosts indicated in purple.”

Reviewer comment: References needs more revision. For example, reference 46 line 373, reference 8 and 53 in table 3.

Author response: We thank the reviewer for their comment to improve the manuscript. We have revised the references as required.

Reviewer comment: H and N in Table S1 should be HA and NA, NEP ….NS

Author response: We appreciate the reviewer’s time and comment regarding the corrections required for Table S1. We have made the corrections as requested and thank the reviewer for highlighting the issue.

Reviewer 2 Report

Rademan's paper focuses on screening Egyptian Rousette bats in South Africa for novel IAV subtypes. Using an RT-PCR approach, the authors identified the recently described bat H9N2.  These results are important because they now document very well that this H9N2 is circulating in bats and is also widespread in South Africa, not just Egypt as previously described.

The paper is well written. There are only few minor issues that the authors might want to address:

1) Did the authors attempt to isolate the virus from bat samples?

2) The reviewer would recommend citing original papers if possible. 

3) Given that the H9N2 virus is circulating in these bats, would it not be feasible to re-examine the bat samples with an RT-PCR approach using specific and highly sensitive primers to estimate the number of infected animals?

Author Response

Reviewer comment: Rademan's paper focuses on screening Egyptian Rousette bats in South Africa for novel IAV subtypes. Using an RT-PCR approach, the authors identified the recently described bat H9N2. These results are important because they now document very well that this H9N2 is circulating in bats and is also widespread in South Africa, not just Egypt as previously described.

Author response: We thank the reviewer for their time, the positive feedback, and constructive suggestions. We have used Grammarly to edit and improve the manuscript’s style and grammar and have addressed the reviewer’s comments and suggestions to the best of our abilities.

The paper is well written. There are only few minor issues that the authors might want to address:

  • Did the authors attempt to isolate the virus from bat samples?

Author response: Unfortunately, we were unable to attempt virus isolation due to low sample volume. Pending future surveillance results to obtain more positive material, virus isolation can be considered together with additional infection studies and characterization.

  • The reviewer would recommend citing original papers if possible.

Author response: We thank the reviewer for the suggestion to improve the manuscript. We have included citations of original papers in the manuscript where applicable as per the reviewer’s request.

  • Given that the H9N2 virus is circulating in these bats, would it not be feasible to re-examine the bat samples with an RT-PCR approach using specific and highly sensitive primers to estimate the number of infected animals?

Author response: Thank you for the excellent suggestion. The primary purpose of the research in the manuscript was to detect any influenza A virus subtypes present and therefore the surveillance approach involved a pan-influenza surveillance assay at the cost of subtype sensitivity. With this manuscript we would like to first release the report of the presence of the virus in Southern Africa (as this was the first detection outside of Egypt) and the sequence data of this bat-H9N2. As part of future surveillance work, we will improve the current RT-PCR approach in respects to the sensitivity of the primers towards the H9N2 bat subtype using the sequence data from the manuscript. We aim to reassess all retrospective samples as well as those collected since, and include investigation of the serological responses of the population to this influenza.

Reviewer 3 Report

This study reported an H9N2 virus originating from bat circulating in South Africa. These findings indicated that influenza A viruses can cross-species and infect bats, suggesting the necessary for the surveillance of bat-origin influenza A viruses. It is a good work and can be published in this journal. I suggest that the authors can try to investigate the antibody positive rate of H9N2 virus in bats in South Africa in the future works.

Author Response

Reviewer comment: This study reported an H9N2 virus originating from bat circulating in South Africa. These findings indicated that influenza A viruses can cross-species and infect bats, suggesting the necessary for the surveillance of bat-origin influenza A viruses. It is a good work and can be published in this journal. I suggest that the authors can try to investigate the antibody positive rate of H9N2 virus in bats in South Africa in the future works.

Author response: We thank the reviewer for their time and positive feedback on the manuscript and for the valuable suggestion. We have used Grammarly to edit and improve the language and grammar of the manuscript. Considering future work, we are aiming to implement serological surveillance with a Luminex-based assay using antigens that are expressed based on the sequences we obtained. This will enable us to further investigate the prevalence and positive rate of the H9N2 bat virus in the Rousettus aegyptiacus population in South Africa.